# The Role of Proximity in Transformational Development: The Case of Resource-Based Cities in China

**Shuo Lu** [1,2], **Wenzhong Zhang** [1], **Jiaming Li** [1,*] **and Renfeng Ma** [3]

1  Institute of Geographic Sciences and Natural Resources Research, Chinese Academy of Sciences, Beijing 100101, China
2  College of Resources and Environment, University of Chinese Academy of Sciences, Beijing 100049, China
3  Department of Geography and Spatial Information Techniques, Ningbo University, Ningbo 315211, China
*  Correspondence: lijm@igsnrr.ac.cn

**Abstract:** Proactive integration into the national city network and the improvement of the level of openness to the outside world is important for the new period to promote the high-quality transformational development of resource-based cities. Based on the gravity model and social network analysis methods, the role of geographical proximity and network proximity (non-spatial proximity) in the high-quality transformation of resource-based cities is systematically analyzed. The impact of geographic proximity, network proximity, and their interactions on the transformation of resource-based cities was also empirically tested by constructing an econometric model. It is found that: (1) The cities with the highest gravitational values and gravitational values among the neighbouring cities of resource-based cities did not change significantly from 2001 to 2019, and cities with better transformation are mostly dominated by RBC-non–RBC combinations, with the gravitational values of neighbouring cities at the middle level. (2) The hierarchy of resource-based cities in both the national organizational network and investment network increased significantly during 2001–2019, but the difference is that the organizational network is centered on Beijing. (3) While there is an inverted U-shaped relationship between geographical proximity and transformation in resource-based cities, there is a linear positive relationship between network investment proximity and transformation. However, there is a substitution and complementary effect between the two, and they work together to promote the high-quality transformation of resource-based cities.

**Keywords:** proximity; resource-based cities (RBCs); point centrality; high-quality transformation

## 1. Introduction

A resource-based city (RBC) is a type of city whose main function is to exploit and process natural resources, such as the minerals and forests in the region [1]. The resource extraction-driven nature of the economy has led to a path-dependent model of urban economic development, and without effective intervention, most industrial and mining cities will go into recession. Since 2007, the Chinese government has initiated the transformation of resource-depleted cities and has achieved some success, but the results are still not significant in some cities. High-quality transformation means transformation with resilience and stability. Transformation has long been a key topic of research in international geography, economics, and political science. At present, the research findings on transformation in RBCs focus on policy [2], environmental regulation [3], industrial structure [4], resource reserves, and local communities [5], in addition to the impact of factors such as investment in education [6] and human capital [7] on urban transformation. As can be seen, much of the research on transformation has focused on the city itself, as well as within the city.

The lower level of external openness of RBCs is an important reason for the path dependence of RBCs. In 2010, the average amount of external investment received by prefecture-level RBCs was only equivalent to 3% of the national average for cities, and after nearly a decade of transformational development, the average amount of external

investment received in 2019 was only 10% of the national average for cities. The lowest amount of external investment was received by the forestry category, 6% of the national average, and the highest amount was received by the non-metal category, which also received only 11% of the national average. If the strategic choice of enhancing the level of opening up of RBCs to the outside world is not accelerated, and does not pay attention to the introduction of capital, advanced technology, management experience, and high-quality talents, it is difficult for RBCs in the period of resource depletion to win a position in the economic cooperation and competition on a large scale, in a wide range of fields, and at a high level [8]. Therefore, the high-quality transformation of RBCs should pay more attention to the impact of production factors from outside the city on the region. As Martinez-Fernandez suggests in his study: a major cause of the contraction and lack of competitiveness in RBCs is the weak linkages of resource-based industries with other networks, which would isolate RBCs from global knowledge networks [9]. RBCs are less capable of industrial renewal and more prone to path-locking problems [10], and external knowledge connections need to be established to undertake economic system renewal. There is no doubt that effectively opening the outside world and grafting new development impulses from outside could be an effective means of creating resource-based pathways [11], but there has been little research exploring the impact of external linkages on high-quality transformation and pathway creation in RBCs [12,13]. Traditional urban studies have only focused on the inner city, while the relationship between the city and other cities on the outside has been largely ignored [14]. In recent years, with the rise of research on urban networks and spatial interactions, proximity has gradually become a hot topic of attention in spatial science, regional economics, and economic geography. The school of proximity dynamics, represented by Torre, Gilly, and Rallet, as well as other EU scholars, represented by Boschma and Balland, have explored the role of multidimensional proximity, including geographical proximity, organizational proximity, and institutional proximity, in cross-regional organizational cooperation and interactive learning.

Based on the existing research results of proximity, this paper attempts to explore the impact mechanisms of proximity on the transformation of RBCs, from the perspective of geographical proximity and network proximity, using the data of enterprise headquarters and branches in China's industrial and commercial enterprise database and the data of inter-city enterprise investment to analyze the spatial connection and network characteristics of 104 RBCs in China and establish a panel model. This paper empirically examines the impact of geographical proximity and network proximity on urban transformation and its mechanisms (Figure 1).

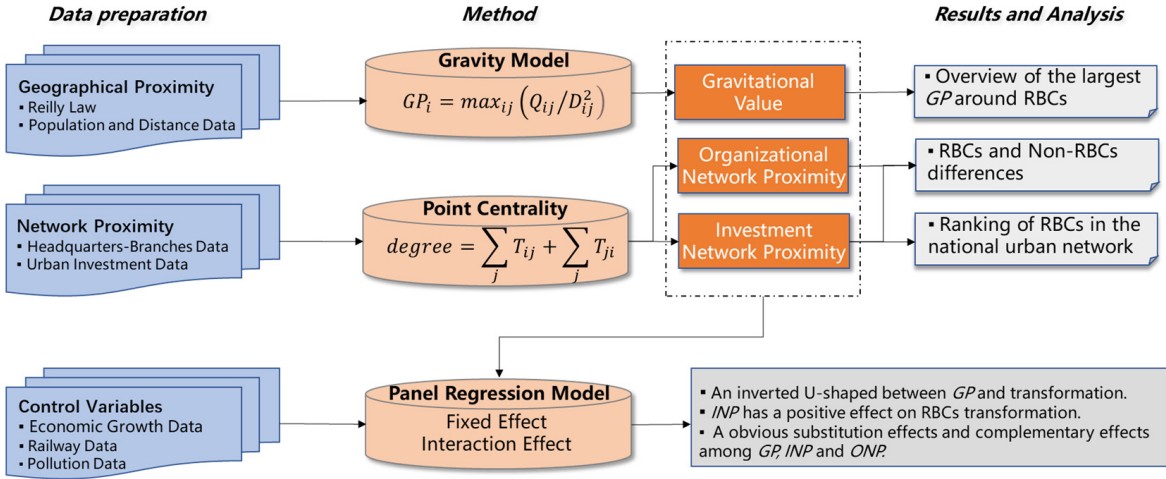

**Figure 1.** Technical framework for this study.

## 2. Theoretical Framework

Since its introduction in the 1990s [15], the concept of path dependence has been widely used in studies in disciplines such as anthropology, history, political science, economics, and management [16], especially in the field of new economic geography, forming a theory of regional growth and development rooted in the concept of path-dependent economic evolution [17], which has become an important power tool for explaining regional economic growth and industrial development patterns. In Martin's model of path-dependent industrial evolution, path dependence is characterized by the persistence and stability of technology, industry, or industrial location pattern, until it is disrupted or expelled by an 'external shock' [18], which is seen as the source of path creation. RBCs are highly dependent on extractive industries, mainly resource extraction, which has led to the consolidation of urban economic development patterns, low resilience of urban economies, and vulnerability to external shocks, such as energy price fluctuations and imbalances in market supply and demand [19,20], eventually leading to a post-boom recession [21]. Resource dependence, furthermore, also leads to educational backwardness and poor investment [22]. This hinders technological progress and sustainable economic growth [23]. Breaking path dependency is not enough to rely on for the self-generated capacity of cities alone, but requires forces from other cities outside to achieve transformation [24]. This non-local strength comes from two main sources, respectively, geographical proximity and network proximity.

Geographical proximity means spatial proximity or physical proximity and refers to the degree of spatial proximity between subjects. Geographically proximate organizations can be closely linked together, facilitating face-to-face interaction between organizations, reducing costs, and increasing the efficiency of information sharing [25]. Traditionally, economic geographers have emphasized the impact of geographical proximity and agglomeration economies in stimulating innovative activity [26], and some scholars have stressed the role of geographical proximity in the production and learning of cluster firms [27]. Even in non-localized networks, where cognitive, organizational, and social proximity are the main drivers of collaboration and innovation, geographical proximity plays an active role [28,29]. On the one hand, geographical proximity can reduce the transport and communication costs for enterprises and establish industrial links with other regions based on original production paths, which helps regions to improve the competitiveness of their own industries [30,31]; on the other hand, geographical proximity can create more offline communication, improve information transparency, reduce transaction costs, and risks, thus promoting transformation [32–34]. However, some scholars argue that excessive geographical proximity tends to result in technological lock-in, spatial lock-in, inflexibility, and the risk of "self-enriching" clusters [35,36].

The French School studies of proximity believed that there is a big difference between network proximity and geographical proximity. Geographical proximity focuses on the spatial distance between actors, while network proximity is related to the closeness and specificity of the actors in an organization. Network proximity is essentially the integration of network theory into the framework of proximity. Laurent, a French scholar, based on the concept of ternary closure, defined network proximity as obtaining the best partners at a lower cost through indirect social ties, weakening the influence of geographical distance and national boundaries. Network theory suggests that the scope of economic activity depends on the structure and position of actors in a network [37]. Participants that are highly embedded in a network can reap greater benefits [38], and those at the core of the network have maximum access to the knowledge resources from the various network members, thereby increasing their own innovation capabilities [39,40]. As fundamental agents of economic development, companies must make use of the integration of knowledge from distant, multi-disciplinary sources, in order to constantly adapt to the changing economic environment [41]. 'Absorptive capacity' is used to describe the ability of firms to acquire and use external knowledge from different fields. This 'absorptive capacity' is particularly important when companies are diversifying or entering new areas [42]. The reason for this is that it provides access to

efficient means of production through neighbourhood networks, which, in turn, provide the factor base and enabling environment for transformation. Van Oort argued that urban networks support a diverse economic environment by expanding market potential, increasing knowledge inputs, and enhancing infrastructure supply [43]. With the development of globalization, information technology, and networking, cities are becoming more and more close to each other, and urban transformation is no longer a 'local transformation', but an important node in the urban network system. If a city or region knows its 'place' in the network, it can create opportunities according to its circumstances. This means that the city or region realizes its economic potential as a consumer, producer, landowner, and investor and is, thus, able to respond positively and innovatively, in order to promote economic development [44]. Many scholars have explored the impact of networks on regional economies and industrial structures, in terms of both business investment and organizational linkages, with Hudson noting the importance of external investment in the transformation of resource-based regions in the northeast of England [45]. The old industrial area of Silesia in Poland started to re-industrialize after the 1990s and applied the program of enhancing organizational network linkages to local transformational development by attracting branches of multinational companies and external funding to create advanced automotive manufacturing clusters through Industry 4.0 strategies [46].

Geographical proximity and network proximity are not independent of each other; they are both complementary and alternative. Malmberg and Maskell argued that there is a significant overlap between geographic and networked forms of proximity and that geographic proximity has a positive impact on the development of networked forms of proximity [47]. For example, when studying the situations influenced by opportunistic behavior, several forms of proximity can work complementarily to maximize benefits [48,49]. At the same time, network proximity can reduce the risk of the negative lock-in faced by geographical proximity [50]. On the other hand, geographical proximity may lead to diminishing returns, due to distance decay, in which case network proximity will replace other forms of proximity. Take the case of two potential partners who are geographically distant from each other, for whom increased network embeddedness is essential for a successful collaboration, as it will be their only source of proximity. Conversely, if they are already close to each other, network proximity will be less important, and geographical proximity will play a decisive role in triggering effective collaboration [51,52]. Thus, geographical proximity becomes the dominant factor in generating the alternative effects of geographical proximity and network proximity (Figure 2).

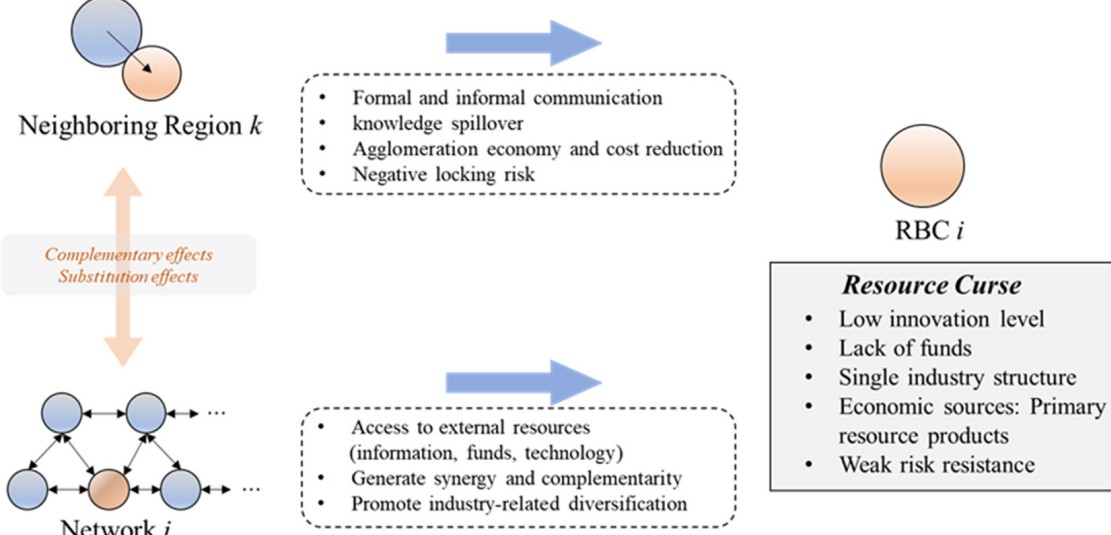

**Figure 2.** Mechanisms of multi-dimensional proximity for RBCs (Self-drawn by the author).

## 3. Data and Methods

### 3.1. Data

In this study, 260 prefecture-level cities (this includes 104 RBCs) in mainland China were selected as the basic research unit, taking into account the availability of data (Table 1). Urban networks are constituted on the basis of urban linkages, and as economic linkages are based on business linkages and the main pathway for quantifying urban networks remains the spatial linkages of regional business organizations. The study is based on Alderson's model of corporate headquarters-branch affiliation [53], where several branches in other cities are considered as several corporate linkages. Totals of 199,249 in 2001, 629,988 in 2010, and 1,320,415 in 2019, respectively, are contained records of investments and headquarters-branches of corporate enterprises, where corporate headquarters refers to the actual office location of the enterprise, rather than the registered location; branch information mainly refers to the city where the enterprise's wholly owned or controlled subsidiaries, branches, offices, and representative offices are located. PM2.5 data were from the PM2.5 historical data website (https://www.aqistudy.cn/historydata/, accessed on 21 April 2020); data on the opening hours of high-speed railway stations were collated from the 12306 website (https://www.12306.cn/index/, accessed on 1 April 2020); data on the number of workers, the number of employees by industry, GDP per capita, and foreign investment data were collected and collated from the China City Statistical Yearbook 2001–2019, as well as provincial and municipal statistical yearbooks.

**Table 1.** Variable Description Statistics.

| Variable | Obs | Mean | Std. Dev. | Min | Max |
|---|---|---|---|---|---|
| RNEE | 312 | 0.89 | 0.12 | 0.43 | 1.00 |
| INP | 312 | 4,607,300.00 | 7,217,703.00 | 6482.48 | 46,000,000.00 |
| ONP | 312 | 1798.97 | 1688.94 | 109.00 | 9595.00 |
| GP | 312 | 2647.80 | 13,027.50 | 23.87 | 147,667.80 |
| FDI | 312 | 22,644.55 | 43,438.44 | 0.00 | 290,822.00 |
| PM2.5 | 312 | 41.24 | 13.86 | 14.58 | 87.47 |
| STAFF | 312 | 32,043.91 | 29,707.47 | 2164.93 | 188,857.00 |
| RGDP | 312 | 262,301.10 | 4,067,694.00 | 2165.00 | 71,900,000.00 |
| HSR | 312 | 0.26 | 0.44 | 0.00 | 1.00 |

### 3.2. Proximity Measure

#### 3.2.1. Geographical Proximity

Based on the concept of dominant flows in transport geography, this paper selects the most attractive city among the neighboring cities of RBCs to characterize geographical proximity (GP) [54,55], considering that the size of the resident population in a municipality is most closely linked to urban development, population movement, and migration, which directly reflect the attractiveness of a city. Therefore, the geographical proximity was calculated based on the city gravity model using the city's resident population as the core indicator. According to Reilly's law, the gravitational value of a surrounding city to the target city is proportional to the population size of the city and inversely proportional to the square of the distance [56]. To sum up, the model was constructed as follows:

$$GP_i = max_{ij}(Q_{ij}/D_{ij}^2) \tag{1}$$

where $GP_i$ is the maximum value of the gravitational force between city $i$ and neighboring city $j$, $Q_{ij}$ is the size of the resident population between city $i$ and neighboring city $j$, and $D_{ij}$ is the geographical distance between city $i$ and neighboring city $j$.

### 3.2.2. Network Proximity

The relative position of participants in a network is important for the establishment of strong network ties between participants; thus, network embeddedness can reflect network proximity [57]. This paper refers to theories related to core-periphery network structures [58], considers the hierarchical position of participants within the network, uses social network analysis tools to assess the position of cities in the network, and measures the position of regions in the network by point centrality [59,60]. The data for quantifying urban networks are mainly based on micro-firm linkages. In highly developed urban systems, where the economic activities of large firms are the main source of interconnections between cities, micro-firms are the most appropriate objects to examine, in order to understand the interactions between cities [61]. Micro-firm linkage data refers to the organizational relationships between the various organizations within a business, as well as the flow of information, capital, senior management, and technical staff [62,63].

This paper uses absolute point centrality to measure organizational network proximity and investment network proximity, respectively, based on micro-firm data on headquarter branches and data on financial flows of investment links between firms, with the basic idea being the number of other points directly connected to a point. If a point is directly connected to multiple points, it implies that the point has a high point centrality. To better measure point centrality in networks of different sizes, Freeman proposed absolute point centrality [64], which is formulated as follows:

$$INP/ONP_i = \sum_j T_{ij} + \sum_j T_{ji} \tag{2}$$

where $INP/ONP_i$ is the point centrality of the city, $T_{ij}$, is the flow data (number of branches and investment size, respectively) for city $j$ where the linked branch is located, the target linkage is city $i$ where the firm is headquartered, and $T_{ij}$ is the value of the reverse linkage. $INP/ONP_i$ characterized 42,643 inter-prefecture headquarters and branch records and 47,358 inter-prefecture business investment data measuring inter-regional organizational network proximity (ONP) and investment network proximity (INP) in 2019, from which, 104 RBCs were selected as research data.

### 3.3. Econometric Model

To explore the mechanism of multidimensional proximity on transformation by comparing the variability of the impact of geographical proximity, organizational network proximity, and investment network proximity on transformation in RBCs, this paper constructs a functional model from a panel data of 104 RBCs in 2001, 2010, and 2019 selected from a sample of 260 studies.

$$RNEE_{it} = \beta_0 + \beta_1 ONP_{it} + \beta_2 INP_{it} + \beta_3 GP_{it} + \beta_4 X_{it} + \mu_i + \varepsilon_{it} \tag{3}$$

where $RNEE_{it}$ indicates the proportion of people employed in non-extractive industries in city $i$ in year $t$. The transformation in RBCs is mainly based on two paths, developing alternative industries and extending the industrial chain, which is manifested by a decreasing proportion of resource extractive industries and an increasing proportion of non-resource extractive industries. With reference to previous studies [1,65], the share of employees in extractive industries is used to characterize the transformation of RBCs, and $ONP_{it}$ indicates the point centrality of business organization of city $i$ in year $t$, characterizing the proximity of the city's organizational network. $INP_{it}$ indicates the point centrality of firm investment in the city $i$ in year $t$, characterizing the proximity of the city's investment network. Referring to Huang's study [66], this paper further selects factors that may affect urban transformation as control variables ($X_{it}$), with the size of the staff employed (STAFF) characterizing the level of human capital. The level of foreign investment utilization (FDI) indicates the degree of internationalization; PM2.5 concentration (PM2.5) indicates the intensity of environmental regulation; GDP per capita (RGDP) indicates the level of re-

gional economic development; and the presence of a high-speed rail station (HSR) is used to measure whether a city is connected to the high-speed rail network, reflecting the level of infrastructure and accessibility. If constant or slow changing over time, this can be controlled for by individual effects $\mu_i$ to avoid missing variables. $\varepsilon_{it}$ indicates the random disturbance term.

Further, to examine the mechanism of the interaction between multidimensional proximity on transformation in RBCs, a squared term and an interaction term were introduced to examine the moderating effect of the interaction between the two on urban transformation, modeled as follows:

$$RNEE_{it} = \beta_0 + \beta_1 GP_{it}^2 + \beta_2 GP_{it} \times INP_{it} + \beta_3 X_{it} + \mu_i + \varepsilon_{it} \tag{4}$$

$$RNEE_{it} = \beta_0 + \beta_1 GP_{it}^2 + \beta_2 GP_{it} \times ONP_{it} + \beta_3 X_{it} + \mu_i + \varepsilon_{it} \tag{5}$$

## 4. Geographical Proximity and Network Proximity Characteristics Analysis

### 4.1. Geographical Proximity of RBCs

The gravitational magnitude of neighboring cities around RBCs is used to reflect geographical proximity, and spatial and temporal trends from 2001–2019 are explored based on the magnitude of the gravitational values of neighboring cities and city types. As can be seen from Figure 3, the gravitational values of the neighboring cities in RBCs were relatively stable overall from 2001 to 2019, with the top ten cities with the highest and lowest gravitational values remaining largely unchanged and the highest city combinations being Ezhou-Huanggang, Xianyang-Xi'an, Jinzhong-Taiyuan, Huangshi-Huanggang, and Fushun-Shenyang; the lowest city combinations were Heihe-Qiqihar, Hulunbeier-Qiqihar, Karamay-Urumqi, Wuwei-Lanzhou, and Wuhai-Shizuishan. This shows a certain inertia in the attractiveness of neighboring cities to RBCs, which is difficult to change in the short term. The spatial layout shows a "core-edge" structure, with cities with high gravitational values being mainly located in central areas such as Shanxi, Anhui, and Hubei and decreasing towards the periphery, with significant differences in the gravitational values between the central and peripheral areas. In terms of gravitational city types, they can be divided into three types: RBC- Provincial Capitals, RBC–RBC, and RBC–non-RBC, with each of the three combination types accounting for nearly 1/3, and it is worth noting that nearly 1/3 of the gravitational cities around RBCs are RBCs. Due to the factors of the industrial attributes of RBCs resulting in their weaker ability to link and drive the surrounding areas, the geographic proximity effect of the resource–resource urban combination is relatively small. Combined with the results of the assessment of the transformation performance of RBCs implemented by the Chinese government, there is a clear bias towards the types of city combinations that are more effective in transformation, with 57.1% of the city combinations being RBC–non-RBC; 28.6% of the city combinations are RBC provincial capitals, while only 14.2% are of the RBC–RBC type. This indicates that RBCs have difficulties in creating collaborative effects and complementarities in inter-city linkages, due to the homogeneity of industry types, which, in turn, prevents them from capturing knowledge spillovers and reducing costs [67]. At the same time, RBCs with good transition assessments have an overall mid-range of neighboring cities' gravitational values, showing that both high and low neighboring cities' gravity values are not conducive to transformation. Too high a gravitational value for neighboring cities may lead to outflows of factors of production, resulting in negative lock-in; too low a gravitational value is not sufficient to generate the externalities of geographical proximity.

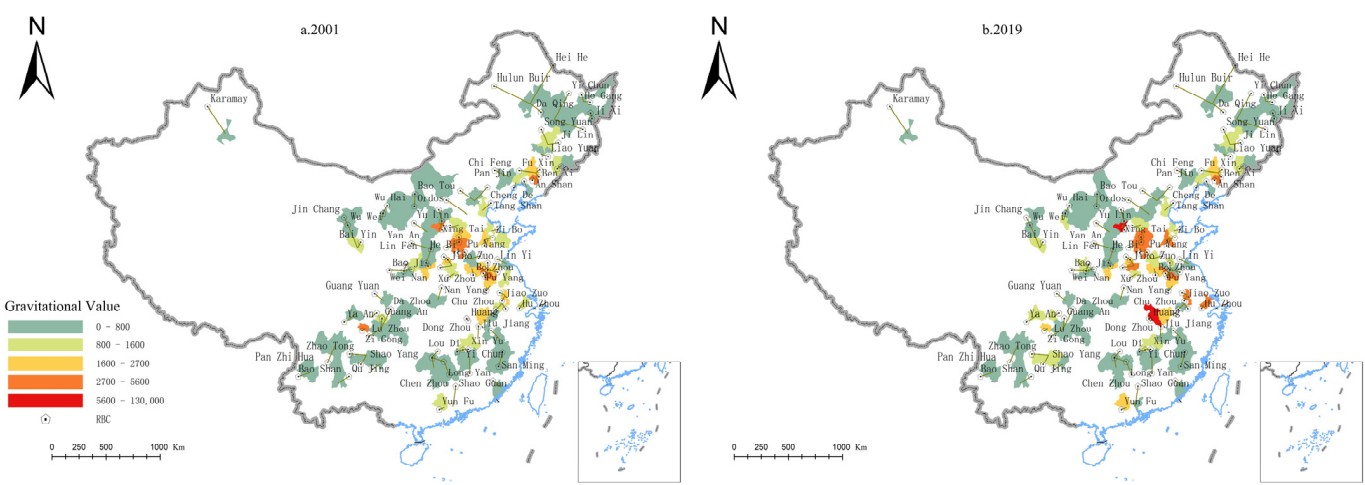

**Figure 3.** Spatial and temporal evolutionary characteristics of geographical proximity.

*4.2. Network Proximity of RBCs*

Point centrality reflects the extensiveness of the linkages of RBCs at the national level, while side link strength is the number of branch structures and total investments received by RBCs from firms in other cities, focusing on the direction and number of linkages and investments received by the RBCs from external firms.

The spatial distribution of cities with high ONP of RBCs has obvious clustering characteristics, and the headquarters of the enterprise branches are concentrated in the Beijing–Tianjin–Hebei region (Figure 4). In terms of the overall pattern of urban firm organizational networks, the hierarchical position of RBCs in the national organizational network increased significantly from 2001 to 2019, with high point centrality of RBCs concentrated in the central and southeastern regions, with obvious clustering characteristics. In terms of organizational linkages, the headquarter layout of enterprise branches in RBCs shows a spatial structure, with Beijing as the center and provincial capitals as the secondary centers. Specifically, in 2001, RBCs had relatively low network rankings in the national network of business organization links. Cities with high point centrality were mainly Xuzhou ($1.59 \times 10^{-3}$), Zibo ($1.43 \times 10^{-3}$), Jining ($1.13 \times 10^{-3}$), and Luoyang ($1.12 \times 10^{-3}$). The headquarters of their companies are located mainly in Beijing. In 2010, the ranking of RBCs in the national organizational network increased, and cities with high point centrality were mainly concentrated around the Beijing–Tianjin–Hebei region, such as Xuzhou ($1.98^* \times 10^{-3}$), Tangshan ($1.64 \times 10^{-3}$), Luoyang ($1.56 \times 10^{-3}$), Zibo ($1.44 \times 10^{-3}$), and Jining ($1.41 \times 10^{-3}$), which are geographically and spatially close to the concentration of headquarters—Beijing—which illustrates that geographical proximity promotes the creation of organizational network proximity. In 2019, RBCs were further ranked in the national organizational network. In addition to the Beijing Ring, they are also concentrated in provinces such as Sichuan, Fujian, and Jiangxi. The cities with high point centrality of RBCs are mainly Xuzhou ($2.92 \times 10^{-3}$), Jining ($2.52 \times 10^{-3}$), Tangshan ($2.47 \times 10^{-3}$), Linyi ($2.47 \times 10^{-3}$), and Zibo ($1.44 \times 10^{-3}$), whose headquarters are concentrated in regional centers such as Shanghai, Fuzhou, and Chengdu, in addition to Beijing.

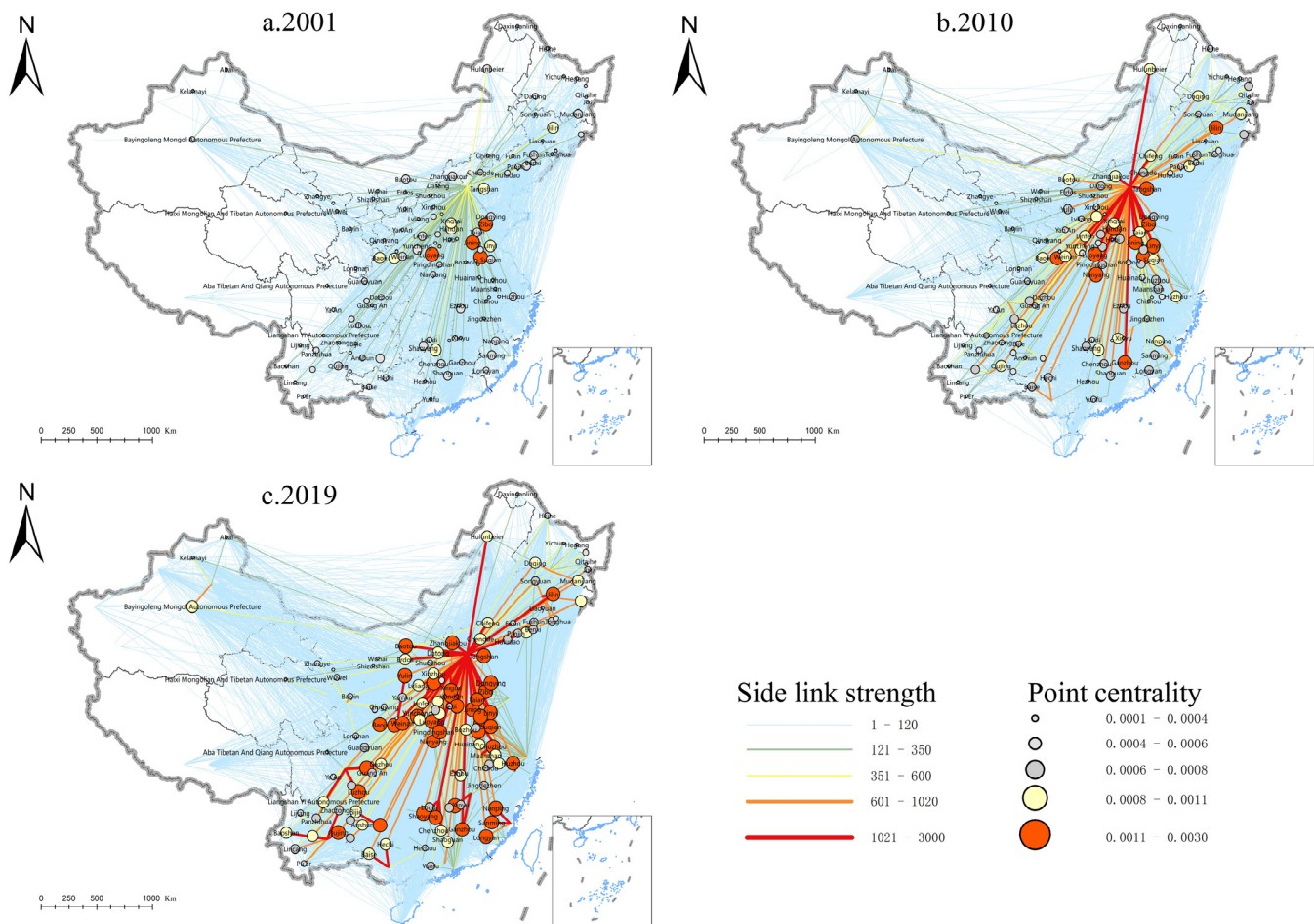

**Figure 4.** Characteristics of the spatial-temporal evolution of the proximity of organizational networks.

The spatial distribution of cities with high proximity to the investment network of RBCs has a clear 'core-edge' character and a diversification from a single source of funding (Figure 5). The spatial distribution of RBCs with high proximity to investment networks has a clear 'core-edge' character and a diversification from a single source of funding (Figure 5). The spatial distribution of RBCs with high INP has a clear 'core-edge' character and a diversification from a single source of funding (Figure 5). In terms of the overall pattern of urban enterprise investment networks, RBCs have been increasing the strength of their linkages in the national investment network from 2001 to 2019, and there is a Matthew effect, meaning that the higher-ranked RBCs in the investment network are more likely to receive external investment, and their spatial pattern is in line with the "core-edge" structure, with higher-ranked cities concentrated in the central and eastern regions and decreasing with distance to the periphery, which is basically consistent with the findings of some urban network studies [68]. In terms of the direction of linkages, RBCs are mainly funded by non-RBCs, and the frequency of investment linkages between RBCs is significantly lower than that between RBCs and non-RBCs. Specifically, in 2001, the ranking of RBCs in the national investment linkage network was generally low, and cities with high INP were mainly Daqing ($0.7 \times 10^{-5}$), Dongying ($0.4 \times 10^{-5}$), Karamay ($0.3 \times 10^{-5}$), and Datong ($0.3 \times 10^{-5}$), and external investment sources were mainly Beijing, showing a single core spatial structure. In 2010, cities with a high INP of RBCs were mainly Erdos ($0.3 \times 10^{-5}$), Tangshan ($0.2 \times 10^{-5}$), Baotou ($0.2 \times 10^{-5}$), and Daqing ($0.5 \times 10^{-5}$), and external sources of investment were mainly Beijing and regional central cities, such as Shanghai and Guangzhou, showing a multi-polar development trend. In 2019, RBCs with high INP are mainly concentrated in the North China Plain, Loess Plateau, and Yunnan-

Guizhou Plateau regions, with external investment sources mainly in the three core regions of Beijing–Tianjin–Hebei, Yangtze River Delta, and Pearl River Delta, as well as regional hub cities, such as Chengdu and Kunming.

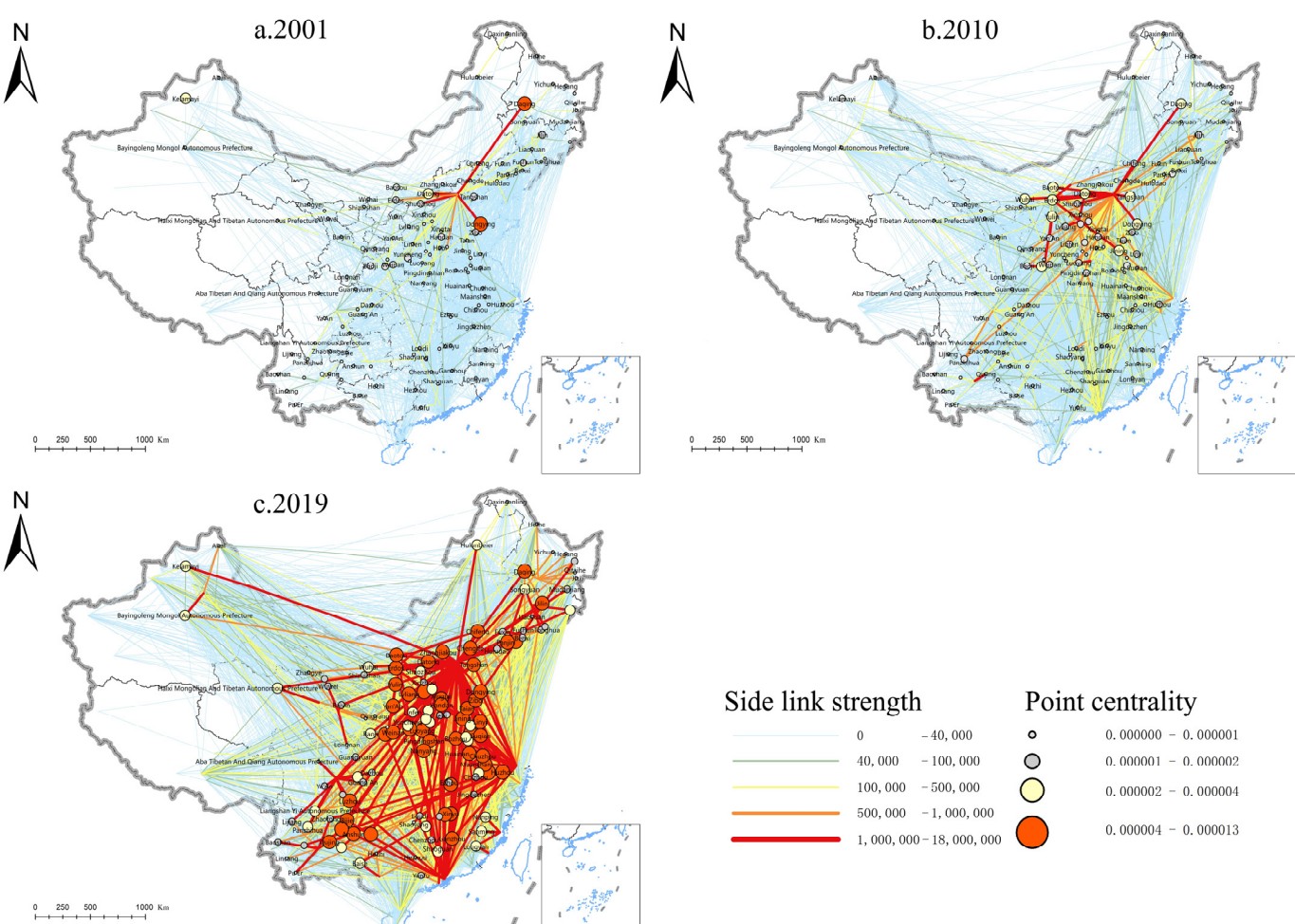

**Figure 5.** Spatial-temporal evolutionary characteristics of the proximity of investment networks.

## 5. Impact of Geographical Proximity and Network Proximity on Transformation

The mechanism of proximity on transformation in RBCs was further explored. The Hausman test shows that the concomitant probability is 0.000, so between the "fixed effect" and "random effect" models, the "fixed effect" model should be used.

There is a significant difference between the impact of geographical proximity and network proximity on industrial transformation under a single proximity dimension perspective. According to the results in Table 2, models 1, 2, and 3 contained moderating variables and main effects, with a significant positive effect of the independent variable investment in network proximity, as seen in model 2 (coefficient = 0.0195, $p < 0.01$). The coefficients of the moderating variables organizational network proximity and geographical proximity on the impact of industrial transformation were negative and insignificant. Models 4, 5, and 6 contained the control variables, moderating variables, and main effects. Model 5 showed that the coefficient on proximity of investment networks was weaker with the inclusion of control variables, but still had a significant positive effect (coefficient = 0.0109, $p < 0.1$), and the size of urban employees had a significant negative effect on industrial transformation (coefficient = 0.0337, $p < 0.01$), to some extent. To a certain extent, the concentration of urban workers in resource-based industries created a 'path lock' effect. The coefficient of the effect of the level of foreign capital utilization on industrial transformation was negative and insignificant (coefficient = −0.0025, $p > 0.1$);

the level of PM2.5 concentration had a significant negative effect on industrial transformation (coefficient = −0.0481, *p* < 0.1), showing that environmental regulation can facilitate the transformation of the industrial structure of RBCs to non-resource-based industries, with a positive and insignificant coefficient for the impact of GDP per capita on industrial transformation (coefficient = 0.0050, *p* > 0. 1). The presence or absence of a high-speed rail station had a significant negative effect on industrial transformation (coefficient = 0.0242, *p* < 0.05), indicating that the construction of high-speed rail stations effectively promotes industrial transformation, as RBCs are better able to access production factors, such as capital and technology, through 'borrowed scale' by being integrated into the high-speed rail transport network.

**Table 2.** Econometric results on the impact of multi-dimensional proximity on urban transformation.

| | Dependent Variable: *RNEE* | | | | | |
|---|---|---|---|---|---|---|
| | **OLS** (1) | **FE** (2) | **RE** (3) | **OLS** (4) | **FE** (5) | **RE** (6) |
| *ln*ONP | 0.0769 ** (−4.35) | −0.0152 (−1.51) | 0.0037 (−0.38) | 0.0833 * (−3.00) | −0.0107 (−0.86) | 0.0229 ** (−2.01) |
| *ln*INP | −0.0293 * (−4.26) | 0.0195 *** (−3.60) | 0.0086 (−1.62) | −0.0263 ** (−4.52) | 0.0109 * (−1.81) | −0.0006 (−0.11) |
| *ln*GP | −0.0032 (−0.97) | −0.0456 (−1.64) | −0.0035 (−0.45) | 0.00624 −2.09 | −0.0468 * (−1.70) | 0.0045 (−0.56) |
| *ln*STAFF | | | | −0.0222 (−0.71) | −0.0337 *** (−2.68) | −0.0147 (−1.37) |
| *ln*FDI | | | | 0.0020 −0.41 | −0.0025 (−1.10) | −0.0022 (−0.98) |
| *ln*PM2.5 | | | | −0.0755 ** (−4.58) | −0.0481 * (−1.76) | −0.0534 ** (−2.37) |
| *ln*RGDP | | | | −0.0224 (−1.49) | 0.0050 −0.78 | −0.0098 (−1.64) |
| HSR | | | | 0.0273 ** −6.11 | 0.0242 ** −2.23 | 0.0232 ** −2.05 |
| _cons | 0.7800 * −9.31 | 1.0230 *** −5.71 | 0.7620 *** −13.40 | 1.1770 * −9.06 | 1.3700 *** −6.23 | 1.0560 *** −10.77 |
| $R^2$ | 0.1165 | 0.1596 | 0.1321 | 0.1862 | 0.2315 | 0.1768 |
| *N* | 312 | 312 | 312 | 312 | 312 | 312 |

*t* statistics in parentheses. * *p* < 0.1, ** *p* < 0.05, *** *p* < 0.01.

The interaction between geographical proximity and network proximity was further explored, in relation to the transformation of industrial structures. Considering that cities around RBCs have an impact on the urban economy in two ways, through the "borrowing scale" to obtain the corresponding positive externality or the "agglomeration shadow" to obtain the corresponding negative externality, this also suggests that there may be a "saturation point" for geographic proximity. This also shows that there may be a 'saturation point' where there is a non-linear relationship between the two. Models 7, 8, and 9, therefore, bring in a quadratic term for geographical proximity and an interaction term for geographical proximity and investment network proximity for regression. The regression results are shown in Table 3. Model 8 shows that the quadratic term of geographical proximity ($GP^2$) had a significant negative effect on the share of non-RBCs industrial structure (coefficient = −0.0043, *p* < 0.05), indicating that geographical proximity and industrial transformation have an inverted "U"-shape relationship. In the first half of the interval where the gravitational value of geographic proximity was relatively small, the two showed a positive correlation, showing that geographic proximity promotes the upgrading of the industrial structure of RBCs. In the second half of the interval, where the gravitational value of geographic proximity was relatively large, the two showed a negative correlation. The implication is that geographical proximity inhibited the transformation and

upgrading of the city's industrial structure. This result confirmed the theoretical frame-work of Figure 1. The effect of geographical proximity on RBCs transformation had a negative lock-in effect, when the gravitational value of neighboring cities was too high, leading to a flow of production factors to the surrounding area and, thus, discouraging the transformation of the industrial structure. At the same time, the interaction term of geographical proximity and investment network proximity showed a significant positive effect (coefficient = 0.0012, $p < 0.05$), which indicated that geographical proximity and investment network proximity can collaborate and complement each other, in order to play a significant positive role in promoting industrial transformation. In particular, in the range of large gravitational values, investment network proximity compensates for the negative lock-in effect of geographical proximity and continues to promote the transformation of the industrial structure of RBCs. Models 10, 11, and 12 were regressed with the quadratic term of geographical proximity and the interaction term of geographical proximity and or-ganizational network proximity. Model 11 showed that the quadratic term of geographical proximity still had a significant negative effect on the share of non-resource-based industry structures (coefficient = −0.0040, $p < 0.1$), which was consistent with the results of model 8. The interaction term between geographical proximity and investment network proximity showed a significant positive effect (coefficient = 0.0020, $p < 0.1$), which also confirmed the complementary effect of geographical proximity and network proximity. As for the control variables, PM2.5 concentration had a significant negative effect on industrial structural transformation in models 8 and 11 (coefficient = −0.0535, $p < 0.05$; coefficient = −0.0655, $p < 0.05$). The size of the workforce had a significant negative effect on the structural trans-formation of the industry (coefficient = −0.0321, $p < 0.01$; coefficient = −0.0310, $p < 0.05$). The presence of a high-speed rail station had a significant positive effect on the structural transformation of the industry (coefficient = 0.0197, $p < 0.1$; coefficient = 0.0223, $p < 0.05$). It can be seen that the regression results are consistent with those in Table 3, indicating that the conclusions in this paper are robust, to a certain extent.

**Table 3.** Econometric results on the impact of multi-dimensional proximity interactions on urban transformation.

| | Dependent Variable: *RNEE* | | | | | |
|---|---|---|---|---|---|---|
| | OLS (7) | FE (8) | RE (9) | OLS (10) | FE (11) | RE (12) |
| *ln*GP$^2$ | −0.0002 | −0.0043 ** | −0.0009 | −0.0023 | −0.0040 * | −0.0012 |
| | (−0.33) | (−1.99) | (−1.13) | (−1.78) | (−1.87) | (−1.51) |
| *ln*GP *ln*INP | 0.0007 | 0.0012 ** | 0.0012 ** | | | |
| | (−0.84) | (−2.15) | (−2.31) | | | |
| *ln*GP *ln*ONP | | | | 0.0061 | 0.0020 * | 0.0032 *** |
| | | | | (−2.33) | (−1.71) | (−3.08) |
| *ln*FDI | 0.0070 | −0.0033 | −0.0016 | 0.0040 | −0.0037 | −0.0027 |
| | (−1.81) | (−1.54) | (−0.79) | (−0.84) | (−1.65) | (−1.26) |
| *ln*PM2.5 | −0.0787 * | −0.0535 ** | −0.0471 ** | −0.0899 ** | −0.0655 ** | −0.0574 *** |
| | (−4.09) | (−2.05) | (−2.18) | (−7.01) | (−2.57) | (−2.69) |
| *ln*STAFF | −0.0034 | −0.0321 *** | −0.0201 * | −0.0169 | −0.0310 ** | −0.0174 * |
| | (−0.22) | (−2.67) | (−1.91) | (−0.67) | (−2.52) | (−1.65) |
| *ln*RGDP | −0.0183 | 0.0019 | −0.0035 | −0.0286 | 0.0018 | −0.008 |
| | (−1.26) | (−0.36) | (−0.69) | (−1.60) | (−0.30) | (−1.48) |
| HSR | 0.0430 ** | 0.0197 * | 0.0220 ** | 0.0152 | 0.0223 ** | 0.0190 * |
| | (−4.34) | (−1.77) | (−2.00) | (−1.76) | (−2.01) | (−1.74) |
| _cons | 1.2430 ** | 1.2730 *** | 1.0940 *** | 1.3330 ** | 1.3270 *** | 1.1520 *** |
| | −7.77 | −8.91 | −12.18 | −6.59 | −9.20 | −12.66 |
| R$^2$ | 0.0942 | 0.2296 | 0.2084 | 0.1423 | 0.2231 | 0.1966 |
| N | 312 | 312 | 312 | 312 | 312 | 312 |

*t* statistics in parentheses. * $p < 0.1$, ** $p < 0.05$, *** $p < 0.01$.

## 6. Conclusions and Discussion

For a long time, transformation studies in the new economic geography have focused on local industrial structures and evolutionary processes, ignoring the impact of proximity.

Focusing on the impact of proximity on transformation, this paper expands the research perspective from single geographical proximity to multidimensional proximity and provides a more comprehensive account of the mechanisms of proximity effects. Using 104 Chinese RBCs as a research sample, the study empirically analyses the mechanisms of geographical proximity, network proximity, and their interactions on transformation in RBCs. The findings of this paper are as follows.

(1) In terms of the changes in the gravitational values of the neighboring cities of RBCs, the gravitational values of the neighboring cities of RBCs were relatively stable overall from 2001 to 2019, with an equal number of RBC provincial capitals, RBC–RBC, and RBC–non-RBC combination types. The cities with good transformation results were mainly RBC–non-RBC combinations, and the gravitational values of their neighboring cities were generally in the middle of the range, which, to a certain extent, also indicates that RBCs are at risk of being adversely affected by the " self-enriching" effect of their cores, if they rely too much on regional centers, such as provincial capitals.

(2) In terms of the national network of cities, the ranking of resource cities in both the national organizational network and the national investment network increased significantly during 2001–2019. The spatial distribution of cities with high proximity to the organizational network of RBCs had obvious clustering characteristics, and the layout of the headquarters of the firms' branches showed a single core characteristic of "Beijing as the core". The spatial distribution of cities with high INP of RBCs had a clear 'core-edge' character and a diversification of funding sources from a single source.

(3) Geographical proximity and transformation are not simply linear, but rather inverted 'U'-shaped. Excessive geographical proximity has a negative lock-in effect and, thus, inhibits industrial transformation, while investment networks proximity has a positive effect on RBCs transformation, so active integration into large regional city networks is conducive to promoting the transformation of RBCs. It is important to emphasize that there are obvious substitution effects and complementary effects among GP, INP, and ONP. In addition, environmental regulation and infrastructure development have a positive effect on transformation in RBCs.

The findings of this study are consistent with the research of several peers. Ye et al., for example, argued that improving relative position in urban networks can enhance the economic resilience of the region [69]. Tsouri's study found that embedding in networks can be effective in accessing needed resources [58]. The transformation dilemma of RBCs is mainly reflected in two aspects. First, the single economic development model of RBCs leads to their vulnerability to external shocks, such as fluctuations in the prices of international energy products and climate change, thus presenting a lack of transformation resilience [8]. Second, the peripheral position of RBCs in both the geographic space and network space makes it difficult to access the production factors needed for quality transformation, such as technology and an advanced production experience [70]. Thus, restricting the transformation to the 'local' level is not an effective solution to the dilemma of transformation faced by resource-based cities, but also requires attention to the transformation of resource-based cities in the 'network'. With RBCs rising in rank in the city network, this can enhance the breadth and depth of the RBCs' external links and improve the ability to capture production factors from the network that match the local region. Additionally, through the integration and matching of external factors and local resources, it has effectively promoted high-quality transformation. This is in line with the theoretical framework of sustainability transitions [71]. Future analysis should focus on the mechanisms by which network proximity influences the transition of RBCs, as well as increasing the exploration of the impact of local institutional factors on network proximity. In this way, the theory of network proximity can be further refined.

This study is an important reference for the policy orientation of the transformation of RBCs. From the theoretical and empirical results, it is clear that active integration into large regional city networks contributes significantly to economic growth and high-quality urban

transformation. High-quality transformation is resilient and stable, and it is clear that integration into the national network of cities and increasing the ranking in the network is a key path to high-quality transformation. Specific policy contributions have been made in two main areas: On the one hand, in the process of collaborative development with neighbouring cities, RBCs should avoid falling into the "agglomeration shadow" by building a friendly business environment, improving infrastructure support, and other initiatives to attract investment, retain enterprises, and support a diversified economic environment. On the other hand, they should further expand the openness to actively integrate into city networks and further enhance the ability of cities to attract businesses and absorb investment. The appropriate use of environmental regulations and other effective means to promote the transformational development of RBCs and narrow the regional development gap should be taken into account.

**Author Contributions:** Conceptualization, W.Z. and J.L.; methodology, S.L.; software, S.L. and J.L.; validation, S.L., J.L. and R.M.; formal analysis, S.L.; investigation, J.L.; resources, W.Z. and J.L.; data curation, S.L.; writing—original draft preparation, S.L., J.L. and R.M.; writing—review and editing, W.Z. and J.L.; visualization, S.L.; supervision, W.Z. All authors have read and agreed to the published version of the manuscript.

**Funding:** This study was funded by the Second Tibetan Plateau Scientific Expedition and Research, grant No. 2019QZKK0406, and the General Program of the National Natural Science Foundation of China (No. 42171178).

**Data Availability Statement:** Not applicable.

**Conflicts of Interest:** The authors declare no conflict of interest.

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
