# Peer review of "The Role of Proximity in Transformational Development: The Case of Resource-Based Cities in China"

_land, doi:10.3390/land11122123_

Round 1

Reviewer 1 Report

The article is academically interesting and for the most part very well written. I have no major complaints, especially as the authors address very well the linking of the research to current trends of theoretical considerations in economic geography. The methodology adopted is clearly outlined and the results are presented in a coherent manner.

I would still suggest focusing the main attention on the "Conclusions and Discussion" chapter. The chapter really only contains conclusions, without any discussion with other scientific studies. I would suggest linking the conclusions to the results of other researchers around the world, so that there is actually room for discussion. If these references are missing, the chapter should not have 'discussion' in its name. There are numerous references to other publications in the article. It might be worth creating a separate "discussion" chapter(?)

Author Response

Thank you for giving us the opportunity to submit a revised draft of the manuscript “The role of proximity in transformational development: The Case of Resource-based Cities in China”.

We appreciate the time and effort that you and the reviewers dedicated to providing feedback on our manuscript and are grateful for the insightful comments on and valuable improvements to our paper. All your and the reviewers’ valuable comments are well taken and the corresponding revisions are colored for highlighting. In particular, we reorganized the literature review and rewrote the discussion to highlight the important value of this study in the Transition theory.

The newly uploaded is described below, divided into two parts:

1) Response Letter to Reviewers;

2) Revised Manuscript.

Reviewer 2 Report

This paper is well written with sound perspective and methodology, and of significance for economic geography and the topic of the restructuring of resource-based cities. I have only two minor concerns that the authors are able to revise them. (1) As the authors explore resource-based cities in China, more references published by Chinese scholars on the topic both from evolutionary and resilience perspectives should be added to improve the visibility of the paper. (2) more theoretical discussions and insight with more references from economic geography should be incorporated into the the conclusion section. It would be great if the Chinese story can contribute to the literature of the restructuring/transformation of resource-based cities in a broader theoretical sense both for the West and China.

Author Response

(The authors gave the same response as above.)

Reviewer 3 Report

The research investigates the role of proximity in transformational development in a Chinese case.

The research is worth considering for the journal and is very appropriate in the way it develops the case study analysis. What needs more work is the theoretical framework as it lacks of the following elements:

- the concepts of walkability and urban regeneration and highlights the connection between proximity, walkability and urban regeneration: https://www.mdpi.com/2071-1050/14/1/457 

- clarifying the relationship proximity-centrality, https://journals.sagepub.com/doi/full/10.1177/2158244020930002

- the connection of proximity with transport accessibility,  https://journals.sagepub.com/doi/pdf/10.1177/2399808317740355

- the connection of proximity with urban planning, https://www.mdpi.com/2071-1050/11/1/31

Please, put the sources of the figures (see Figure 2. Mechanisms of multi-dimensional proximity for RBCs.)

These are the reasons to require a new version of the paper.

Author Response

Dear  Reviewer: 

Thank you for giving us the opportunity to submit a revised draft of the manuscript “The role of proximity in transformational development: The Case of Resource-based Cities in China”.

We appreciate the time and effort that you and the reviewers dedicated to providing feedback on our manuscript and are grateful for the insightful comments on and valuable improvements to our paper. All your and the reviewers’ valuable comments are well taken and the corresponding revisions are colored for highlighting. In particular, we reorganized the literature review and rewrote the discussion to highlight the important value of this study in the Transition theory.

The newly uploaded is described below, divided into two parts:

1) Response Letter to Reviewers;

2) Revised Manuscript.

Round 2

Reviewer 3 Report

The paper is ready for publication.